# Biodiversity measures of a grassland plant-pollinator community are resilient to the introduction of honey bees (*Apis mellifera*)

**Sydney H. Worthy**●*, **John H. Acorn, Carol M. Frost**

Department of Renewable Resources, University of Alberta, Edmonton, Alberta, Canada

* worthy@ualberta.ca

**Data Availability Statement:** Data available in Dryad: https://doi.org/10.5061/dryad.1vhhmgqzg. Voucher specimens of all insects have been deposited in the E.H. Strickland Entomological

## Abstract

The prairies of Canada support a diversity of insect pollinators that contribute pollination services to flowering crops and wild plants. Habitat loss and use of managed pollinators has increased conservation concerns for wild pollinators, as mounting evidence suggests that honey bees (*Apis mellifera*) may reduce their diversity and abundance. Plant-pollinator community analyses often omit non-bee pollinators, which can be valuable contributors to pollination services. Here, we experimentally introduced honey bees to examine how their abundance affects the species richness, diversity, abundance, species composition, interaction richness, and interaction diversity of all wild pollinators, and of four higher taxa separately. We identified all insect pollinators and analyzed how honey bee abundance affected the above biodiversity metrics, controlling for flower abundance and flower species richness. Even with high honey bee densities, there was no change to any of these variables, except that beetle species diversity increased. All other taxa had no significant relationship to honey bee abundance. Considering the widespread use of managed honey bees, the effect they have on wild pollinators should be firmly established. Our results suggest that honey bees have little to no short-term impact on the wild pollinator community or its interactions with plants in this native grassland.

## Introduction

Honey bees (*Apis mellifera*, Linnaeus, 1758) are widely used for honey production and crop pollination, and often in regions where they are not native. Recent studies have suggested that high honey bee densities may have negative effects on wild pollinator abundance and diversity through competition for floral resources [1–3]. Insect pollinators are diverse and are important for both wild plant and crop pollination [4], performing a variety of valuable ecosystem functions [5]. While bees are the most effective pollinators for many plant species, non-bee insects can also be effective pollinators by the sheer number of visits they provide [4, 6]. A high taxonomic and functional diversity of wild pollinator species can promote the resilience of the community; that is, the capacity to absorb disturbance and change while retaining the same function and structure [7–9].

Museum, University of Alberta, Edmonton, Canada,
with accession numbers UASM430001-
UASM430601.

**Funding:** S.H.W. was provided funding by the
University of Alberta's Rangeland Research
Institute and awarded funding by the Alberta
Conservation Association. There were no award or
grant numbers associated with these grants. RRI -
https://rri.ualberta.ca/ ACA - https://www.ab-
conservation.com/ The funders had no role in
study design, data collection and analysis, decision
to publish, or preparation of the manuscript.

**Competing interests:** The authors have declared
that no competing interests exist.

With similar body sizes and foraging habits, bumble bees may be especially subject to honey bee competition [10], with high honey bee abundance negatively affecting bumble bee fitness [11, 12], and negatively correlating with bumble bee abundance [13, 14] and foraging behaviour [15]. Honey bees may also impact solitary wild bees by outperforming them in pollen collection [16], and affecting their flower visitation and reproductive output [17]. Honey bees may impact non-bee insect pollinators, such as flies, beetles, and butterflies, which are valuable pollinators alongside bees [4, 18–20], in the same way. However, few studies have explored honey bee impact on non-bee insect pollinators. Lindström et al. [21] found that honey bees depressed the densities of some flies in oilseed rape, despite a high density of floral resources, and Ropars et al. [22] noted a negative correlation between honey bee abundance and flower visitation rates for large-bodied bees and beetles, though there was no negative correlation with other groups (hover flies or butterflies).

In a recent review, Iwasaki & Hogendoorn [23] found that, in studies regarding resource competition, 68% reported negative effects of honey bees and other introduced bees. Competitive effects on wild pollinators were lower in studies where honey bees were within their native range [3]. Herbertsson et al. [24] found that honey bee abundance suppressed bumble bee density in homogenous landscapes, but not in heterogenous landscapes, though Zink [25] found the opposite. Overall, the effect of honey bees seems to vary temporally [15, 26, 27], regionally, and by landscape, and therefore appears to be context-dependent.

Honey bees may also affect plant- pollinator interaction richness (the number of different plant-insect species pair interactions) and interaction diversity (the number and relative frequency of plant-insect species pair interactions). Honey bees, interacting with many plants, may displace interactions from native pollinators, decreasing the community's interaction richness and diversity [28]. In a previous study in the same system, we investigated whether varying honey bee abundances affected total plant-pollinator network structure, as well as the structure of just the wild pollinator-plant interactions [29]. We found that although total network structure changed as honey bee abundance increased (because honey bees themselves added so many interactions), the network structure of the wild pollinator-plant interactions did not change [29]. However, network structural metrics do not necessarily capture biodiversity changes [30], especially for non-dominant species, and interaction richness and diversity may be lost even while interaction network structural metrics do not change significantly. As such, both species and interaction diversity metrics may be more sensitive to honey bee density than network metrics.

Temporal variation in honey bee effects on wild pollinators may be driven by temporal variation in honey bee colony size. The population of a Canadian honey bee colony peaks in mid-summer, reaching between 50,000–80,000 individuals per hive [31]. In 2019 (the year of this study), there were approximately 314,800 managed honey bee colonies in Alberta [32], suggesting that a minimum of 3.15 billion honey bees are added to the ecosystem annually, with substantial increases in mid-summer. Little work has been done on temporal effects of honey bee colony size on wild pollinators, with some exceptions. Walther-Hellwig et al. [15] found a higher density of honey bees at midday, associated with the depression of bumble bee density. Additionally, Wignall et al. [27] found that exploitative competition increased between bumble bees and honey bees during July and August. Honey bees may have strong competitive effects on wild pollinator species whose emergence coincides with peak honey bee colony size, like some leafcutter bees [33].

As it is imperative to investigate any potential effect of honey bees to ensure wild pollinator conservation, our objectives were to: i) determine whether honey bee abundance impacts the species richness, diversity, abundance, or species composition of wild pollinators; ii) determine whether honey bee abundance impacts the richness and diversity of plant- pollinator

interactions; iii) determine whether honey bee abundance affects the richness, diversity or abundance of certain higher taxa (bees, beetles, flies, and butterflies) more than others; and iv) determine whether effects of honey bees are greater when honey bee colony sizes are largest, and whether this differs for different higher taxa.

## Methods

### Experimental design

The methods described here are partially the same as Worthy et al. [29], where additional site details can be found. Worthy et al. [29] reports the effects of honey bee abundance on functional characteristics (network interactions) of the plant-pollinator community, whereas this study reports on the effects of honey bee abundance on the biodiversity of the wild pollinator community, in part by utilizing pan-trapped data (see below), which Worthy et al. [29] does not. Briefly, we arranged three clusters of honey bee hives in mixedgrass prairie rangeland, set at least 3 km apart, on the Mattheis Research Ranch near Brooks, Alberta, Canada (Fig 1; GPS locations in S1 Table). The hive clusters contained 48, 32, and 16 hives respectively, imitating the range of honey bee densities around small-scale commercial apiaries. The hives were assessed bi-weekly by the apiarist for health and productivity throughout the season. By late summer, all hives grew significantly and had 4 to 6 supers added per hive.

Since 1970, this ranch has used light cattle stocking rates (approximately 0.7 animal unit months ha$^{-1}$) [34]. The two most popular flowering crops in this region, canola and alfalfa, were not grown within 17 and 8 km, respectively, of the ranch boundary line in 2019. Additionally, the Eastern Irrigation District, owner of the land surrounding the Mattheis Research Ranch, reported no known apiaries within 19 km of the ranch's boundary line within the study year or the previous year. Eighteen 30 x 2 m transects at 100 m, 500 m, and 5000+ m

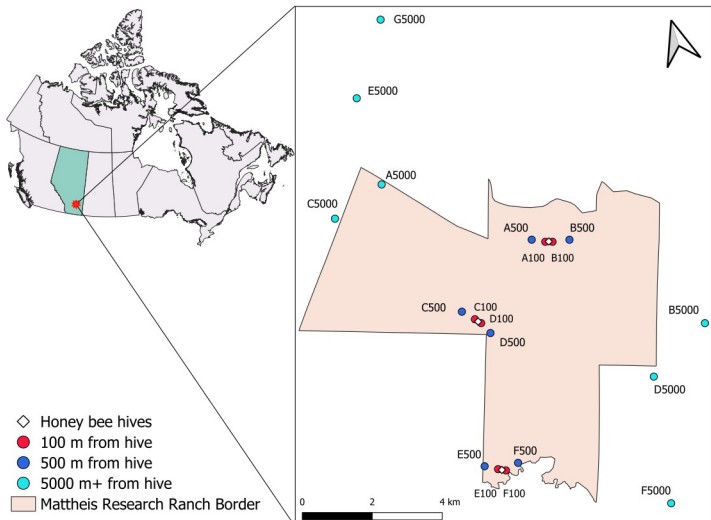

**Fig 1. Map of the research location, Mattheis Ranch, in southern Alberta, Canada.** Locations of the sampling transects (circles) and honey bee hives (diamonds). Each hive location contained a cluster of hives (northernmost: 48 hives, central: 32 hives, southernmost: 16 hives). The transect G5000 was established when sampling was discontinued at F5000 (after July 9th, 2019). Sampling transects outside the Mattheis Ranch were on land managed by the Eastern Irrigation District. Reprinted from Worthy et al. [29] under a CC BY license, with permission from PLOS ONE, original copyright 2023. Map created in QGIS [36] with shapefiles from Natural Resources Canada under an Open Government license [37].

distances were established on either side of each cluster of honey bee hives for a total of six replicates (Fig 1). These distances were selected because Hagler et al. [35] found that recovery of marked honey bees decreased exponentially with distance from the hive, with the steepest decrease in number of marked bees recovered occurring between 100 and 500 m from the hive, and very low numbers of honey bees detected 2500+ m from the hive. We expected our 5000+ m transects to have little to no honey bee visitation.

Two observers visited each transect and surveyed insect flower visitation almost once per week, weather permitting, between May 28 and August 28, 2019, for a total of 10 collection rounds. The first transect at the beginning of each day was selected on a rotating schedule, and subsequent transects were sampled along a route of highest efficiency. During some collection rounds, some transects could not be sampled due to cattle presence or a lack of flowers, resulting in different sampling amounts for each transect (S1 Table). Transects were visited only on warm ($\geq$ 15°C), sunny days with wind speeds under 40 km/h to improve the chance of pollinator activity [38, 39]. We measured wind speed with a Brunton Sherpa (Riverton, Wy, USA). Pollinator sampling occurred between 9:30 am and 5:00 pm, when flower visits are most frequent [40].

Two people observed the transects for 30 minutes each for a total of 60 person-minutes per transect per collection round (4200 total collection minutes). All insects that visibly contacted the anthers or stigma of open flowers were collected with a hand net and placed in labelled individual vials, frozen, and identified to species in the lab (the "hand-caught" dataset, also used in an analysis of network structure in Worthy et al. [29]; Fig 2a). Some specimens (5%) were identified to morphospecies. When insects observed landing on flowers were missed, they were recorded, but these data were excluded because the species-level identifications for these observations were heavily biased toward larger-bodied pollinators. When large volumes of insects were captured, sampling was paused via a timer for processing and then resumed. All identifications were completed using a dissecting microscope, taxonomic literature, expert taxonomist help, and comparison with reference specimens (S2 Table). Following pollinator sampling, on each sampling round, all flowering plants on the transect were identified and their flowers were counted (S2 Table). Occasionally, flowers were not present on the marked transect, so we moved the transect $\leq$ 10 m from its original location to reach flowers near the original transect demarcation. Moving the transect did not alter the distance to the hives.

Pan traps were placed to take a second set of measurements of diversity, abundance, and species composition at each transect that were more independent of the local plant assemblage than the same measurements from the hand-caught dataset (the "pan-trapped dataset", Fig 2b). Pan traps target flower-visiting insects, and tend to provide more complete sampling of the flower-visitor community than does hand-netting [42]. Two blue, white, and yellow pan traps were placed at 10-m intervals on the ground along either side of each transect for a total of six pan traps per transect. Each pan trap had a few drops of Dawn dish detergent in water [43, 44]. Pan traps were retrieved no later than 55 hours after placement, with exposed time recorded to the nearest quarter hour.

After July 9, 2019, sampling was halted at one 5000 m transect ("F5000") after many honey bees were observed (presumably from a feral or unregistered hive). Since this replicate was intended to be a low-honey bee density transect, a new 5000 m transect ("G5000") was selected approximately 8000 m away from the northern 48-hive cluster (Fig 1). If unknown, additional honey bee colonies had contributed interactions to our transects, we should have detected them during sampling such as this. Similarly, one transect ("E100") had few flowers, such that pollinator sampling could not be done after the third collection round; however, pan traps were still collected. We treated these transects as distinct site replicates but controlled

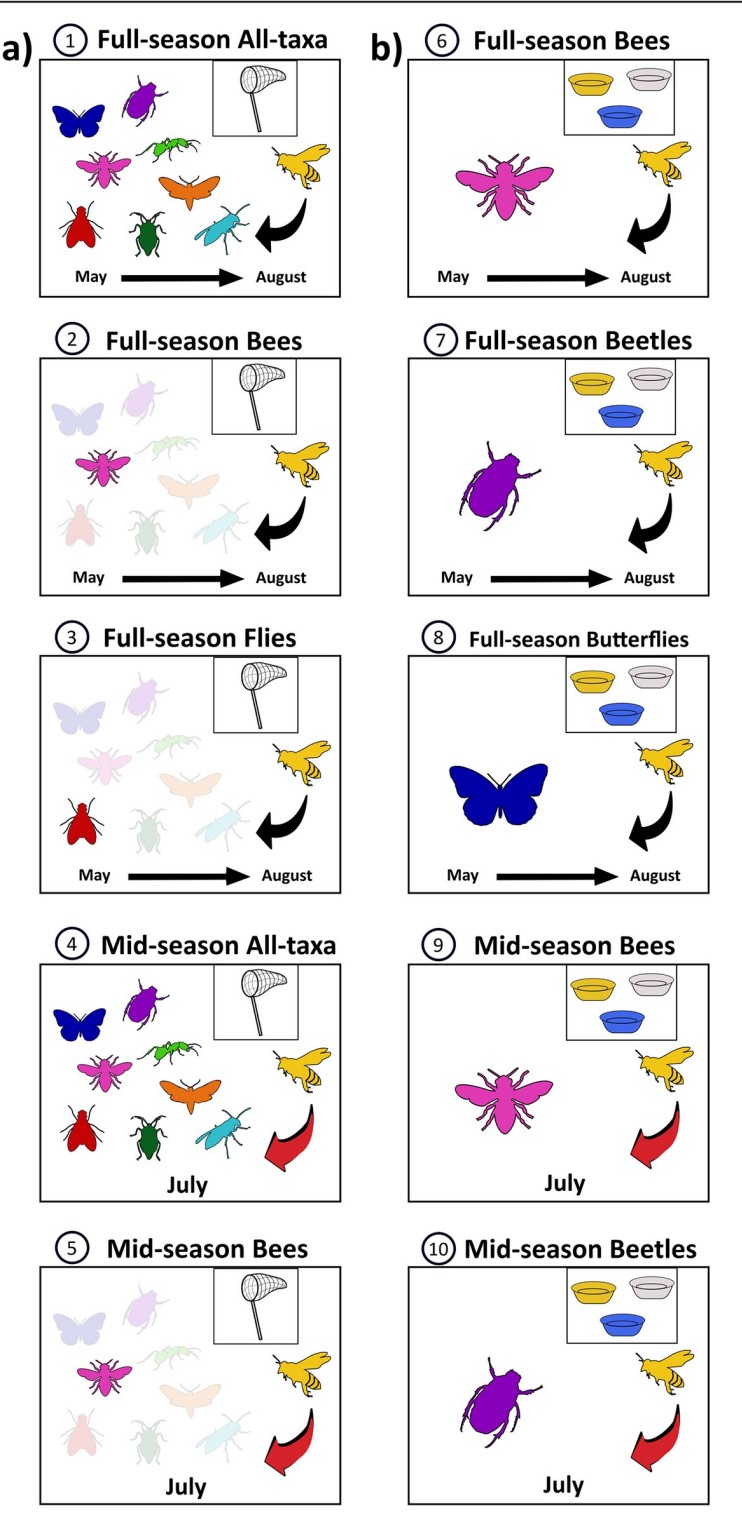

**Fig 2. Visual representation of the ten datasets tested in this study.** The ten datasets (1–10) were derived from the hand-caught dataset (a) and the pan-trapped datasets (b). Full-season datasets were completed over all collection rounds (May 28th-August 28th), and mid-season datasets were only during three collection rounds (July 8th-July 31st). Colours indicate different taxonomic groups of insect pollinators: yellow, honey bees; pink, non-Apis bees; navy blue, butterflies; purple, beetles; lime green, ants; red, flies; forest green, true bugs; orange, moths; sky blue, wasps. Black

arrows indicate the hypothesized effect of honey bees on insect flower visitors, and red arrows indicate a stronger effect expected during the mid-season window when honey bee population was highest. Insect images are from SVG SILH [41].

statistically for the lower number of collections at these by including number of collections as a covariate in our multiple regression models (see below and S1 Table).

This study involved the collection of plant and invertebrate animal materials from the Mattheis Research Ranch, a research area owned by the University of Alberta, and the Eastern Irrigation District (EID) in Southern Alberta. Permission was granted by both and did not require a field permit number.

## Pollinator diversity metrics

To assess the effects of honey bee abundance on different taxa and at different times of the season, we created several separate datasets from the hand-caught dataset (Fig 2a): 1) full-season all-taxa, 2) full-season non-*Apis* bees, 3) full-season flies, 4) mid-season all-taxa (a reduced dataset including only three collection rounds from July 8th to July 31st), and 5) mid-season non-*Apis* bees. Additionally, we created several separate datasets from the pan-trapped dataset (Fig 2b): 6) full-season non-*Apis* bees, 7) full-season beetles, 8) full-season butterflies, 9) mid-season non-*Apis* bees, and 10) mid-season beetles. Some mid-season higher taxa groups, such as flies and butterflies, had too few individuals and were excluded from analysis.

For each dataset, we pooled the data over the full-season or mid-season period respectively, and then calculated pollinator rarefied species richness (a measure of the expected species richness for a given sample size), with sample size held constant across all sites, where the sample size was equal to the number of individuals collected at the site with the fewest collected individuals ("species richness") (vegan package, [45, 46]). When calculating rarefied species richness in the full-season all-taxa (1) and the pan-trapped full-season non-*Apis* bees dataset (6), we retained the transects "E100" and "G5000" in all calculations. However, a cut-off point was established that excluded transects (or higher taxa) without enough individuals from rarefaction analysis, to increase the accuracy of richness estimates. Ultimately, 10 and 15 individuals were chosen as arbitrary cut off values for the hand-caught and pan-trapped datasets, respectively, which meant that some transects were removed from analysis in all datasets except 1 and 6 (see S3 Table). Additionally, for only the full-season all-taxa dataset (1), we calculated rarefied interaction richness, the expected number of different interactions, holding the total number of interactions sampled constant (hereafter "interaction richness").

We used second order (Simpson) Hill numbers [47, 48] to assess pollinator species diversity ("species diversity"). We also calculated second order Hill numbers for the pairwise interactions between pollinators and plants ("interaction diversity"). We measured pollinator abundance as the number of individual pollinators. Pollinator species composition ("species composition") was recorded as a matrix of species abundances by transect and pollinator species.

## Statistical analyses

We used linear regression to test the effect of honey bee abundance on species richness, species diversity, and abundance, with the abundance of honey bees from each transect in the full-season all-taxa hand-caught dataset, divided by collection effort for that transect, as the predictor variable. We used this predictor variable, rather than using transect distance to hives, or honey bee abundance from the pan-trapped dataset, as in [49], and for the same reasons as those

authors gave: the number of honey bees in the pan traps did not appear to reflect the number of honey bees observed in flight, or on flowers while sampling, as well as the number of hand-caught honey bees did, as found by [50]. Furthermore, neither hand-caught (S1 Fig) nor pan-trapped (S2 Fig) honey bee abundance followed a pattern of decreasing abundance as hive distance increased, meaning that distance from bee hives likely did not always reflect potential competitive effect of honey bees. However, hand-caught honey bee abundance was higher than pan-trapped honey bee abundance, and was therefore the predictor variable of interest in all statistical models, including those modeling pan-trapped abundance and diversity.

We used simple linear regression (SLR) to test the effect of honey bee abundance on each response variable for all datasets (1 through 10), and for interaction richness and interaction diversity in the full-season all-taxa dataset (1). We ran a SLR for each, with honey bee abundance as the only predictor variable. As some transects were close together (200 m), we created a map of the standardized residuals of each SLR for which honey bee abundance was significant at $\alpha = 0.05$, to inspect whether a special correlation structure was necessary to account for spatial autocorrelation (S3 Fig). Generalized least squares (GLS) mixed models were then run with different correlation structures (no correlation, corEcp, corGaus, corSpher, corLin, corRatio) for each response variable (nlme package, [51]). The AICc (Akaike's Information Criterion, corrected for small sample size) values of each model were calculated, and the model with the lowest AICc value (or any simpler model within 2 AICc points of the model with the lowest AICc; [52]) was selected (MuMin package, [53]). The best models for each response variable did not include special correlation structures, so SLRs rather than GLS models were run, except for beetle abundance and non-*Apis* bee species richness in datasets 9 and 10, respectively (S4 Fig).

If honey bee abundance was significant in the first model, we ran a secondary multiple regression (MR) for that response variable, with all predictor variables included in the full model. This was to determine whether the significant effect of honey bee abundance that appeared in the SLR disappeared when we included all the variables that might affect our response variables and be correlated with honey bee abundance. If honey bee abundance was still a significant predictor in these models, this would suggest that honey bee abundance affected the response variable independent of any correlations between honey bee abundance and these other variables [54]. We used the same model selection approach described above to select the predictor variable(s) that best explained that response variable. Each continuous predictor variable was standardized by subtracting the variable's mean from each observed value and dividing by the variable's standard deviation [55]. We ran the full model, and all possible simpler models, for a total of 21 linear models per response variable (stats package, [56]), and selected the final model using AICc as described above. Although our approach controls for any effects of multicollinearity on our interpretation by allowing comparison of SLR and MR results for the effect of honey bee abundance on each response variable for which the relationship was significant in the SLR, we also calculated variance inflation factors for the full season analysis and the midseason analysis, which we present in S4 Table.

We examined the assumptions of normality for each response variable by running a Shapiro-Wilk test on the residuals of the best models. We assessed homogeneity of variance by examining plots of fitted values versus residuals. If the assumptions were not met, we log-transformed that response variable and repeated model selection on the transformed response variable, after which if assumptions were still not met, we applied a Box-Cox transformation (MASS package, [57]). In some cases, transformation did not improve normality or variances, in which case we present the results but do not interpret them.

We used permutational multivariate analysis of variance (PERMANOVA) to assess whether pollinator species composition was affected by honey bee abundance, using both the Bray-

Curtis and Jaccard distance metrics (vegan package, [45]). Bray-Curtis distance takes into account species' relative abundances, whereas Jaccard just takes into account presence/absence, such that using both metrics allows understanding of whether any changes to species composition were to species presence/absence or relative abundances. We used non-metric multidimensional scaling (NMDS) to visualize patterns in species composition (ecodist package, [47]). We ran the NMDS with the function metaMDS which runs the function monoMDS until two similar configurations with minimized stress are found [45]. All analyses were completed using R version 3.2.4. [56]. For a list of datasets and the response variables calculated for each, see S5 Table.

To assess whether the effects of honey bee abundance were more pronounced in the mid-season, we plotted honey bee abundance from the hand-caught dataset pooled across the season and divided it into three roughly equal-length periods: "early", "mid", and "late", based on natural breaks in the hand-caught abundances of honey bees (S5 Fig). The mid-season period was reduced to July 8th to July 31st, 2019. Since honey bee abundance was highest in the mid-season window, we tested the effects of honey bee abundance on all the above response variables for only the mid-season datasets (datasets 4, 5, 9, and 10). We ran a SLR for each response variable with honey bee abundance as the predictor variable, and if honey bee abundance was significant, we ran a secondary MR for each response variable, with all predictor variables included as described above. We then ran a PERMANOVA for each dataset, testing for effects of honey bee abundance on species composition. This analysis interpreted 59 separate models, so to maintain Type I Error at 0.05, we used a Bonferroni-Holm correction (at $P$ = 0.00085). However, all $P$-values are presented to allow a less conservative consideration of uncorrected $P$-values.

## Results

A total of 281 pollinator species and 37 plant species were identified from 1,815 interactions in the hand-caught dataset. Bees were identified to 73 species (half of the number of species of bees found in pan traps, and 25% of the known bee number of species in the region [58]). Butterflies were identified to 15 species (approximately half of the number of species in pan traps, and 27% of the known number of species in the region, based on Bird et al. [59] excluding riparian species, but including occasional migrants). Beetles and ants were identified to 14 and 11 species, respectively (11.8% of the number of ant species estimates in the region [60]). Detailed pollinator identifications from the hand-caught dataset are listed in S6 Table. Honey bees were the most common pollinator (16% of the hand-caught dataset interactions) and western snowberry (*Symphoricarpos occidentalis)* was the most commonly visited flowering plant (20% of the hand-caught dataset interactions). Of the 1,815 recorded interactions, 654 were unique species pair interactions. Of these, 425 were only observed once.

From the pan-trapped dataset, 11,437 bee, beetle, and butterfly specimens were identified to 240 species. Flower-count data revealed a total of 46 flowering plant species, approximately 32% of the flowering plant species in the entire grassland region of Alberta [61]. All plant species are listed in S7 Table. A total of 6,655 bees were identified to 147 species, approximately half of the known bee species of southern Alberta [58]. A total of 2515 butterflies were identified to 28 species, representing half of the known butterfly species of the region [59]. A total of 2,278 beetles were identified to 65 species. Total pan-trapped hours were 6032.25 pooled across the 2019 season.

Overall, bees were predominantly small-bodied sweat bees (Halictidae: *Lasioglossum* spp.). The most abundant butterfly was the plains skipper (Hesperiidae: *Hesperia assiniboia*), and the most abundant beetle was the rust-coloured blister beetle (Meloidae: *Epicauta ferruginea*). The

most abundant wasp species were *Stenodynerus anormis* (Vespidae) and *Ectemnius rufifemur* (Crabronidae). The most abundant fly species was *Eristalis stipator* (Syrphidae) and the most abundant ant species was *Formica lasioides* (Formicidae). All pan trap identifications are listed in S8 Table. Rarefaction curves showed that the hand-caught full-season all-taxa dataset (1) and mid-season all-taxa dataset (4) did not plateau, whereas the curves for the pan-trapped higher taxa datasets (6, 7, 8) and the mid-season pan-trapped higher taxa datasets (9, 10) began to flatten but did not fully plateau (S6 Fig). For all datasets, the 95% confidence intervals for the rarefaction curves overlapped for all distances from hives, though overlap was lowest for the mid-season hand-caught dataset, especially for the 100 m and 500 m distances (S6b Fig). Only in this dataset was there a trend toward pollinator richness being highest at the 5000 m distance and lowest at the 100 m distance.

### Effect of honey bees on pollinator diversity metrics

Honey bee abundance (plotted per transect in S1 Fig) did not significantly affect any metric, after Bonferroni-Holm correction, for all higher taxa, trapping methods, including in the mid-season datasets (Tables 1 and 2), except that there was a significant increase in beetle species diversity with honey bee abundance, after controlling for flower abundance, flower species diversity, collection effort, as well as the interactions between honey bee abundance and flower abundance, flower species diversity, and collection effort (Table 2g; Fig 3). In general, the lowest (though non-significant) *P*-values were associated with positive relationships between honey bee abundance and the diversity metrics, rather than the predicted negative associations (Tables 1 and 2).

Species composition was not significantly affected by honey bee abundance for any dataset, and the NMDS for the full-season all-taxa dataset showed no substantial separation of pollinator species composition by honey bee treatment (Fig 4). Stress levels for the NMDS plot were 0.20, indicating a good fit for the model, where values of zero indicate a perfect fit between data and ordination distances. For the PERMANOVAs run with both Bray-Curtis and Jaccard dissimilarity, the lowest *P*-values (though non-significant after Bonferroni-Holm correction) were for effects of honey bee abundance on mid-season hand-caught Non-*Apis* bee species composition and full-season pan-trapped Non-*Apis* bee species composition (Tables 3 and 4), and for full-season pan-trapped butterfly species composition with Jaccard dissimilarity (Table 4).

### Discussion

Honey bees appeared to have little to no impact on the wild pollinator community or the number and frequency of their interactions with plants over one growing season. Sporadic bee keeping has occurred in and around the study area in the past, but the Eastern Irrigation District reported no known apiaries within 19 km of the ranch's boundary line within the study year or the previous year. Additionally, the law requires apiaries to annually register their hive (s) to the Provincial Apiculturist. This suggests that the diverse insect pollinator community of Alberta's grasslands can maintain or quickly recover the state of the ecosystem, despite the addition of managed honey bees. Although many studies have indicated negative impacts of honey bees on wild pollinators [11, 16, 17, 20–22, 62, 63], our study is in agreement with those that found either a neutral or positive effect on wild pollinators [13, 22, 24, 64, 65].

At the transect level, we saw no significant effects of increasing honey bee abundance on species richness in any dataset. When transects were pooled by treatment (as in Valido et al. [20]), these trends held, except for mid-season hand-caught species richness, in which there was a trend toward lower species richness with decreasing distance to hives (S6b Fig). Only

**Table 1. Results of the simple linear regression (SLR) models with the lowest AICc values for each response variable, where honey bee abundance as the only predictor variable.**

| Response variable | Confidence intervals (95%) | | Adjusted R$^2$ | Regression coefficient | t-value | *P*-value |
|---|---|---|---|---|---|---|
| **Hand-caught dataset** | | | | | | |
| **a) Full-season all-taxa (Dataset 1)** | | | | | | |
| Species richness | 0.08 | 0.94 | 0.23 | 0.5103 | 2.50 | **0.0231** |
| log(Species diversity) | 0.04 | 0.60 | 0.21 | 0.3180 | 2.42 | **0.0272** |
| Abundance | -0.31 | 0.69 | -0.02 | 0.1916 | 0.81 | 0.4320 |
| Interaction richness | -1.04 | 0.94 | -0.06 | 0.4702 | -0.10 | 0.9180 |
| log(Interaction diversity) | -0.62 | 0.14 | 0.04 | -0.2386 | -1.33 | 0.2020 |
| **b) Full-season Bees (Dataset 2)** | | | | | | |
| Non-*Apis* bee species richness | -0.18 | 1.09 | 0.08 | 0.4531 | 1.52 | 0.1490 |
| Non-*Apis* bee species diversity | -0.67 | 3.07 | 0.05 | 1.2024 | 1.37 | 0.1900 |
| Non-*Apis* bee abundance | -0.33 | 0.63 | -0.04 | 0.1460 | 0.65 | 0.5270 |
| **c) Full-season Flies (Dataset 3)** | | | | | | |
| Fly species richness* | -0.40 | 16.05 | 0.19 | 7.8220 | 2.06 | 0.0606 |
| Fly species diversity | -0.33 | 3.30 | 0.13 | 1.4838 | 1.76 | 0.1010 |
| log(Fly abundance) | -0.61 | 0.77 | -0.15 | 0.0794 | 0.28 | 0.7880 |
| **d) Mid-season all-taxa (Dataset 4)** | | | | | | |
| Species richness* | -4.41 | 18.39 | 0.04 | 6.9890 | 1.31 | 0.2110 |
| log(Species diversity) | -0.25 | 0.55 | -0.03 | 0.1461 | 0.78 | 0.4460 |
| Abundance | -0.37 | 0.46 | -0.06 | 0.0462 | 0.24 | 0.8160 |
| **e) Mid-season Bees (Dataset 5)** | | | | | | |
| Non-*Apis* bee species richness | -1.12 | 0.50 | -0.02 | -0.3127 | -0.83 | 0.4220 |
| Non-*Apis* bee species diversity | -0.46 | 0.24 | -0.04 | -0.1109 | -0.68 | 0.5090 |
| Non-*Apis* bee abundance | -7.11 | 4.25 | -0.04 | -1.4310 | -0.53 | 0.6010 |
| **Pan-trapped dataset** | | | | | | |
| **f) Full-season Bees (Dataset 6)** | | | | | | |
| Non-*Apis* bee species richness* | -5.44 | 15.4 | 0.01 | 5.0020 | 1.01 | 0.3260 |
| Non-*Apis* bee species diversity | -1.31 | 3.51 | 0.03 | 1.1010 | 0.96 | 0.3490 |
| Non-*Apis* bee abundance | -15.02 | 123.21 | 0.09 | 54.1000 | 1.65 | 0.1170 |
| **g) Full-season Beetles (Dataset 7)** | | | | | | |
| Beetle species richness | 0.25 | 1.45 | 0.32 | 0.8454 | 2.99 | **0.0087** |
| Beetle species diversity | 0.57 | 1.80 | 0.48 | 1.1841 | 4.09 | **0.0009** |
| Beetle abundance (*CorGaus*) | -18.74 | 33.63 | 0.23 | 7.4453 | 0.56 | 0.5846 |
| **h) Full-season Butterflies (Dataset 8)** | | | | | | |
| Butterfly species richness | 0.02 | 1.33 | 0.18 | 0.6740 | 2.19 | **0.0437** |
| Butterfly species diversity | 0.06 | 1.09 | 0.22 | 0.5740 | 2.38 | **0.0303** |
| Butterfly abundance | -48.00 | 7.79 | 0.68 | -20.1000 | -1.52 | 0.1470 |
| **i) Mid-season Bees (Dataset 9)** | | | | | | |
| Non-*Apis* bee species richness (*CorRatio*) | 0.21 | 0.93 | 0.17 | 0.5715 | 3.12 | **0.0071** |
| Non-*Apis* bee species diversity | -3.52 | 1,07 | 0.02 | -1.2250 | -1.14 | 0.2730 |
| Non-*Apis* bee abundance | -41.13 | 20.42 | -0.03 | -10.3600 | -0.72 | 0.4840 |
| **j) Mid-season Beetles (10)** | | | | | | |
| Beetle species richness | -0.14 | 1.28 | 0.12 | 0.5699 | 1.72 | 0.1070 |
| Beetle species diversity | -0.12 | 1.13 | 0.12 | 0.5022 | 1.73 | 0.1060 |
| Beetle abundance* | -0.66 | 0.29 | -0.02 | -0.1885 | -0.85 | 0.4100 |

No P-values were significant with Bonferroni-Holm correction. Bolded P-values denote P ≤ 0.05, which meant that the second (MR) model with all predictor variables was run (Table 2). Asterisk (*) denotes Box-Cox transformation, and other transformations are listed with the response variable. Hand-caught datasets 1–5 denoted by a-e, and pan-trapped datasets 6–10 denoted by f-j.

**Table 2.** Results of the multiple regression (MR) models with the lowest AICc value for each response variable, where the full model contained honey bee abundance, flower abundance, flower species richness, number of collection rounds, the interactions between honey bee abundance and flower abundance, and between honey bee abundance and flower species richness, as predictor variables.

| Response variable | Predictor variables retained in final model | Confidence intervals (95%) | | Adjusted R$^2$ | Partial regression coefficient | t-value | P-value |
|---|---|---|---|---|---|---|---|
| **Hand-caught datasets** | | | | | | | |
| **a) Full-season all-taxa (Dataset 1)** | | | | | | | |
| Rarefied species richness | flower richness | 0.47 | 1.09 | 0.60 | 0.7774 | 5.30 | P < 0.001 |
| log(Species diversity) | flower richness | 0.35 | 0.70 | 0.69 | 0.5287 | 6.40 | P < 0.001 |
| **Pan-trapped datasets** | | | | | | | |
| **g) Full-season Beetles (Dataset 7)** | | | | | | | |
| Beetle species richness | honey bee abundance | 0.25 | 1.45 | 0.32 | 0.8454 | 2.99 | 0.0087 |
| Beetle species diversity | honey bee abundance | 0.98 | 2.37 | 0.60 | 1.6763 | 5.16 | **0.0001** |
| | flower richness | -1.52 | -0.97 | | -0.8067 | -2.42 | 0.0285 |
| **h) Full-season Butterflies (Dataset 8)** | | | | | | | |
| Butterfly species richness | flower richness | 0.29 | 1.49 | 0.34 | 0.8870 | 3.13 | 0.0065 |
| Butterfly species diversity | honey bee abundance | 0.06 | 1.09 | 0.22 | 0.5740 | 2.38 | 0.0303 |
| **i) Mid-season Bees (Dataset 9)** | | | | | | | |
| Non-*Apis* bee species richness (*CorRatio*) | honey bee abundance | 0.21 | 0.93 | 0.17 | 0.5715 | 3.12 | 0.0071 |

Bolded P-values denote significance with Bonferroni-Holm correction. Asterisk (*) denotes Box-Cox transformation, and other transformations are listed with the response variable.

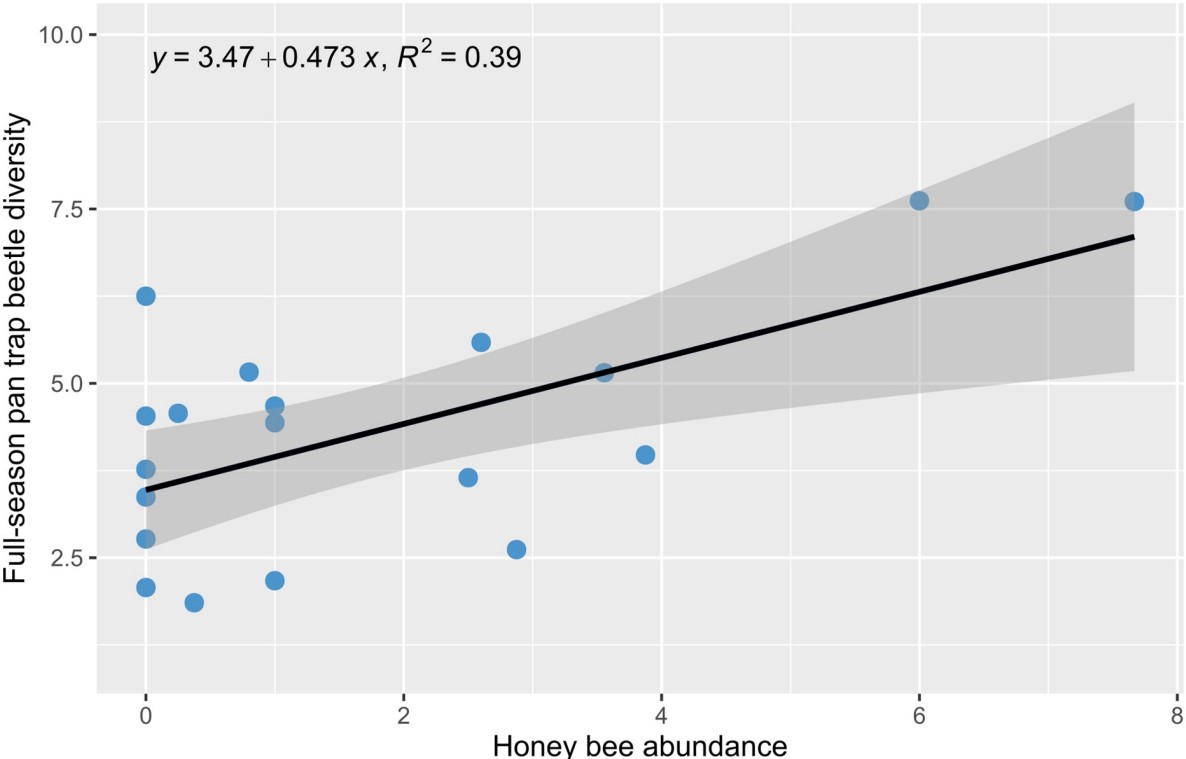

$$y = 3.47 + 0.473\,x,\ R^2 = 0.39$$

**Fig 3. Scatterplot of honey bee abundance against full-season pan-trapped beetle diversity.** The y axis indicates full-season pan-trapped beetle diversity and x axis indicates honey bee abundance. The regression line is fitted from the SLR model (Table 2g).

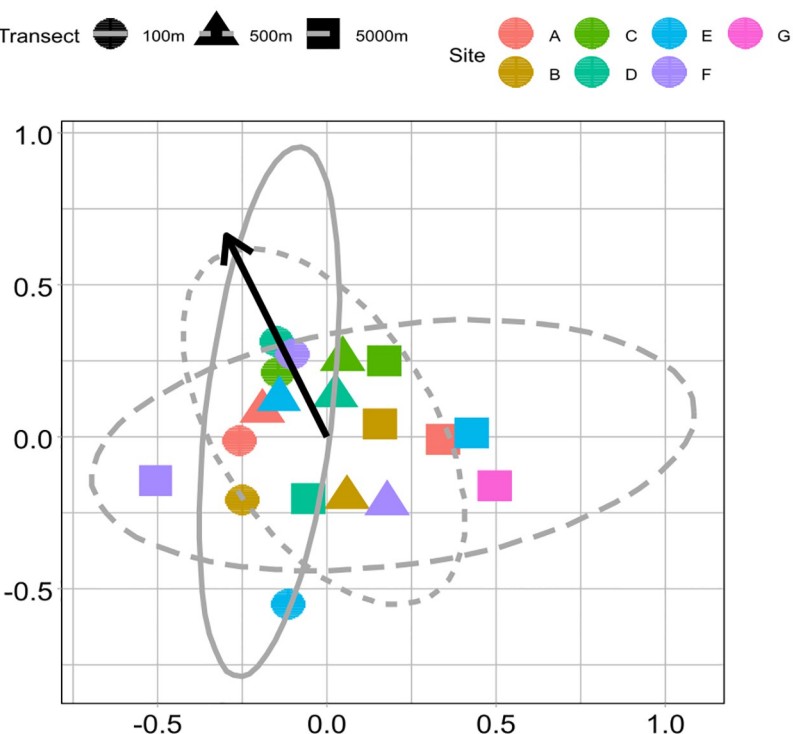

**Fig 4. NMDS Plot of flower visitor species composition per transect.** Each point represents a transect, with closer-together points on the plot having more similar flower visitor species composition. Ellipses represent 95% confidence and shapes indicate treatment (100 m, 500 m, or 5000 m distances from a hive). The stress score was 0.20. Bolded vector indicates Apis mellifera abundance.

one significant positive relationship was found, between honey bee abundance and the species diversity of pan-trapped beetles over the full-season (Tables 1 and 2; Fig 3). It may be that when honey bees are present, there are other, unmeasured variables related to floral quality that also attract other wild pollinators, such as volatiles, flower density, stem height, corolla length, nectar quality, or pollen availability [66, 67]. Though not significant after correction for multiple comparisons, honey bee abundance also had a positive relationship to the richness and diversity of other taxa. Interestingly, there was no effect of honey bee abundance on inter-action richness or interaction diversity, despite Valido et al. [20] finding a negative effect of honey bees on interaction richness. However, the high number of one-time interactions may indicate that a large number of interactions were undetected by sampling; caution should therefore be used when interpreting these interaction richness and interaction diversity results.

Due to low abundances of some taxa, only flies, butterflies, bees, and beetles could be examined separately for effects of honey bees on their diversity metrics. Still, this is one of the few studies to examine the effects of honey bees on non-bee taxa. Lindström et al. [21] found a negative relationship between honey bee densities and overall wild pollinator densities; however Ropars et al. [22] found that visitation rates of wild solitary bees, flies, and butterflies were related to plant species richness rather than honey bee density. In our study, when analyzing the second MR model for each response variable in each dataset, which included all predictor variables, flower species richness appeared to have the strongest positive relationship with overall pollinator species richness, species diversity, and fly species diversity (Table 2). Overall,

**Table 3. Results for PERMANOVA tests for effects of honey bees on pollinator species composition with Bray-Curtis dissimilarity.**

| | Response variable | Sum of squares | Pseudo-F | R$^2$ | *P*-value |
|---|---|---|---|---|---|
| **Hand-caught datasets** | | | | | |
| | **a) Full-season all-taxa (Dataset 1)** | | | | |
| | Species composition | 0.0401 | 0.6463 | 0.0366 | 0.4690 |
| | **b) Full-season Bees (Dataset 2)** | | | | |
| | Non-*Apis* bee species composition | 0.0628 | 0.7170 | 0.0405 | 0.4520 |
| | **c) Full-season Flies (Dataset 3)** | | | | |
| | Fly species composition | 0.1349 | 1.0950 | 0.0605 | 0.3470 |
| | **d) Mid-season all-taxa (Dataset 4)** | | | | |
| | Species composition | 0.0782 | 1.3707 | 0.0789 | 0.2650 |
| | **e) Mid-season Bees (Dataset 5)** | | | | |
| | Non-*Apis* bee species composition | 0.2813 | 3.5867 | 0.1930 | 0.0380 |
| **Pan-trapped datasets** | | | | | |
| | **f) Full-season Bees (Dataset 6)** | | | | |
| | Non-*Apis* bee species composition | 0.1612 | 3.0765 | 0.1532 | 0.0800 |
| | **g) Full-season Beetles (Dataset 7)** | | | | |
| | Beetle species composition | 0.0794 | 0.6682 | 0.0378 | 0.4860 |
| | **h) Full-season Butterflies (Dataset 8)** | | | | |
| | Butterfly species composition | 0.1391 | 1.9684 | 0.1038 | 0.1240 |
| | **i) Mid-season Bees (Dataset 9)** | | | | |
| | Non-*Apis* bee species composition | 0.0611 | 0.9139 | 0.0540 | 0.3570 |
| | **j) Mid-season Beetles (Dataset 10)** | | | | |
| | Beetle species composition | 0.2049 | 1.7215 | 0.0971 | 0.1880 |

Honey bee abundance was the only predictor variable. Bolded P-values denote significance with Bonferroni-Holm correction.

though with the caveat that we only tested over one season, our results suggest that the presence of honey bees alone, even at high densities and in peak season, did not negatively impact the diversity or abundance of wild pollinators, or the diversity of their plant-pollinator interactions.

As Alberta's grasslands support a variety of flowering plants, it may be that the niches of honey bees and wild pollinators did not totally overlap, as observed by Herbertsson et al. [24]. Honey bees (and other large-bodied pollinators, such as *Bombus* spp.) were not observed to visit small-flowered and small-stemmed plants, such as *Campanula rotundifolia*, which attracted many smaller pollinators. Honey bees also appeared to prefer introduced species, such as *Astragalus cicer* and *Melilotus* spp., as in Urbanowicz & Muñiz [68]. Therefore, honey bees may have avoided resources that small-bodied pollinators could utilize.

A caveat to our study is that only 20–50% of the richness of this region was sampled (depending on higher taxon). However, pan trap and hand-net sampling hours rivaled or surpassed those of other studies that found a negative effect of honey bee abundance on plant-pollinator networks [20, 69], which suggests that the sampling effort in this study should have been sufficient to detect any effects of honey bees, if they existed. Over a longer period of time, some effects of honey bee abundance may have become detectable, as other variables, such as rainfall or temperature, may have affected competition between honey bees and wild pollinators. Additionally, while we could not encompass the entire growing season for every forb species, the majority of flowering in this region historically occurs from May through August [70].

**Table 4. Results for PERMANOVA tests for effects of honey bees on pollinator species composition with Jaccard dissimilarity.**

| | Response variable | Sum of squares | Pseudo-F | R² | *P*-value |
|---|---|---|---|---|---|
| **Hand-caught datasets** | | | | | |
| | **a) Full-season all-taxa (Dataset 1)** | | | | |
| | Species composition | 0.0561 | 0.5033 | 0.0288 | 0.6470 |
| | **b) Full-season Bees (Dataset 2)** | | | | |
| | Non-*Apis* bee species composition | 0.0899 | 0.6204 | 0.0352 | 0.5960 |
| | **c) Full-season Flies (Dataset 3)** | | | | |
| | Fly species composition | 0.1739 | 0.9161 | 0.0511 | 0.4300 |
| | **d) Mid-season all-taxa (Dataset 4)** | | | | |
| | Species composition | 0.1409 | 1.2865 | 0.0744 | 0.2660 |
| | **e) Mid-season Bees (Dataset 5)** | | | | |
| | Non-*Apis* bee species composition | 0.3792 | 2.8226 | 0.1584 | 0.0450 |
| **Pan-trapped datasets** | | | | | |
| | **f) Full-season Bees (Dataset 6)** | | | | |
| | Non-*Apis* bee species composition | 0.2522 | 2.5984 | 0.1326 | 0.0550 |
| | **g) Full-season Beetles (Dataset 7)** | | | | |
| | Beetle species composition | 0.1555 | 0.8649 | 0.0484 | 0.4370 |
| | **h) Full-season Butterflies (Dataset 8)** | | | | |
| | Butterfly species composition | 0.2559 | 2.4356 | 0.1253 | 0.0990 |
| | **i) Mid-season Bees (Dataset 9)** | | | | |
| | Non-*Apis* bee species composition | 0.1655 | 1.4208 | 0.0816 | 0.2420 |
| | **j) Full-season Beetles (Dataset 10)** | | | | |
| | Beetle species composition | 0.3095 | 1.7453 | 0.0984 | 0.1800 |

Honey bee abundance was the only predictor variable. Bolded P-values denote significance with Bonferroni-Holm correction. Blocks of similarly shaded rows demarcate separate datasets.

## Conclusions

The effect of honey bees on wild pollinator communities is contentious, and this study provides further evidence indicating that negative effects may be context-dependent [23]. Additionally, our findings suggest that reducing managed honey bee densities may not be an effective, or necessary, action for wild pollinator conservation in this region. Future research is needed to determine whether honey bees can coexist in the long run with wild pollinators in agriculturally-dominant regions where native grassland is becoming more scarce, and whether there are potential negative effects of disease transmission from honey bees.

## Supporting information

**S1 Table. Longitude and latitude for each hive location and transect, and collection effort at each transect.** The northernmost, central, and southernmost hive locations are listed as Bee48, Bee32, and Bee16 respectively, each associated number indicating the number of hives. Each transect is indicated by its treatment (100 m, 500 m, or 5000 m distance from a hive location). Letters indicate each replicate (See Fig 1). G5000 indicates the new position for F5000 that was moved mid-season. Reprinted from Worthy et al. [29] under a CC BY license, with permission from PLOS ONE, original copyright 2023.
(DOCX)

**S2 Table. List of references and resources used in species identifications.** Full citations are listed in the References section of the Supporting Information. Reprinted from Worthy et al. [29] under a CC BY license, with permission from PLOS ONE, original copyright 2023.
(DOCX)

**S3 Table. Number of individual insect pollinators, excluding honey bees, that were caught per transect for datasets 1 to 10 (see Fig 2).** Datasets 1 to 5 are derived from the hand-caught dataset, and datasets 6 to 10 are derived from the pan-trapped dataset.
(DOCX)

**S4 Table. Variance inflation factors calculated for both the full-season and mid-season to assess multicollinearity between variables.**
(DOCX)

**S5 Table. List of datasets and metrics calculated for each dataset (See Fig 2).** Each metric was used as the response variable in a statistical test of the effect of honey bee abundance on that variable. Analyses for some higher taxa from the mid-season all-taxa dataset could not be completed when abundances were too low (flies, butterflies, ants, moths, true bugs). Interaction richness and interaction diversity were only analyzed for the full-season all-taxa dataset.
(DOCX)

**S6 Table. Identifications of insect pollinators to species-level or morphospecies level from the hand-caught dataset.** Morphospecies identifications are listed by "[Genus] sp. #". Some species could not be differentiated between genera, and so both genera are listed along with the epithet "sp". Specimens listed beside "cf" (confer, meaning compare with) are specimens that were damaged or for which taxonomic keys are insufficient, and these were compared to other specimens to determine identification. Numbers of each (morpho)species are given for each distance from hives, despite the fact that we used honey bee abundance, rather than distance from hive as the predictor variable in our analyses. Reprinted from Worthy et al. [29] under a CC BY license, with permission from PLOS ONE, original copyright 2023.
(DOCX)

**S7 Table. Identifications of flowering species from each distance from honey bee hives to species level.** Reprinted from Worthy et al. [29] under a CC BY license, with permission from PLOS ONE, original copyright 2023.
(DOCX)

**S8 Table. Identifications of insect pollinators to species-level or morphospecies level from the pan-trapped dataset.** Morphospecies identifications are listed by "[Genus] sp. #". Some species could not be differentiated between genera, and so both genera are listed along with the epithet "sp". Specimens listed beside "cf" (confer, meaning compare with) are specimens that were damaged or for which taxonomic keys are insufficient, and these were compared to other specimens to determine identification. Numbers of each (morpho)species are given for each distance from hives, despite the fact that we used honey bee abundance, rather than distance from hive as the predictor variable in our analyses.
(DOCX)

**S1 Fig. Abundance of honey bees caught visiting flowers (Dataset 1).** Honey bees were pooled across the full season per transect, with transects ordered by increasing honey bee abundance, and coloured by distance from bee hives. In the transect names, 100 indicates 100 m, 500 indicates 500 m, and 5000 indicates 5000 m distances from hives. The Eastern Irrigation District, owner of the land surrounding the Mattheis Research Ranch, reported no known

apiaries within 19 km of the ranch's boundary line. All commercial or hobbyist apiaries are required by law to register their hive(s) annually to the Provincial Apiculturist. Reprinted from Worthy et al. [29] under a CC BY license, with permission from PLOS ONE, original copyright 2023.
(TIF)

**S2 Fig. Honey bee abundance in pan traps, pooled across the full season per transect, with transects ordered by increasing honey bee abundance, and coloured by distance from nearest bee hives.** In the transect names, 100 indicates 100 m, 500 indicates 500 m, and 5000 indicates 5000 m distances from hives. Reprinted from Worthy et al. [29] under a CC BY license, with permission from PLOS ONE, original copyright 2023.
(TIF)

**S3 Fig. Visual representation of the spatial autocorrelation of residuals for each response variable against honey bee abundance for response variables that did not need a special correlation structure.** Circles indicate the size of the residual for each transect (smaller circles = better model fit). Colour indicates the sign of the residual; blue shows values lower than 0 and red values higher than 0. In this figure, if close together transects have similarly sized and coloured residuals, that suggests that there is spatial autocorrelation in that response variable. Reprinted from Worthy et al. [29] under a CC BY license, with permission from PLOS ONE, original copyright 2023.
(TIF)

**S4 Fig. Visual representation of the spatial autocorrelation of the residuals for the two response variables that did require special correlation structures.** Both models required rational quadratic special correlation structures. Circles indicate the size of the residual for each transect in the model without any special correlation structure (smaller circles = better model fit). Colour indicates the sign of the residual; blue shows values lower than 0 and red values higher than 0. In this figure, if close together transects have similarly sized and coloured residuals, that suggests that there is spatial autocorrelation in that response variable. Reprinted from Worthy et al. [29] under a CC BY license, with permission from PLOS ONE, original copyright 2023.
(TIF)

**S5 Fig. Total abundance of honey bees caught visiting flowers pooled across all transects.** The entire season's collection rounds were split into three groups: collection rounds 1–4 represented "early" season (May 28th-July 7th), 5–7 represented "mid" season (July 8th-July 31st), and 8–10 represented "late" season (August 1st-August 28th). Reprinted from Worthy et al [29] under a CC BY license, with permission from PLOS ONE, original copyright 2023.
(TIF)

**S6 Fig.** Species rarefaction curves for a) full-season all-taxa dataset (1), b) mid-season all-taxa dataset (4), c) pan-trapped higher taxa dataset (6,7, and 8), and d) mid-season pan-trapped higher taxa dataset (9 and 10), where the richness of each higher taxon in the pan-trapped dataset (bees, beetles, butterflies) were pooled together. These curves plot the average number of species obtained from repeated random re-sampling of the number of individuals given on the x-axis. Transects were pooled by distance from hives, as indicated by green (100 m), red (500 m), and blue (5000 m). Confidence intervals indicate 95%. Figures were generated with the iNEXT package.
(TIF)

## Acknowledgments

We thank Brittany Wingert, Irene Jimenez Roncancio, James Glasier, Greg Pohl, Lincoln Best, and the University of Calgary Museum of Zoology for their assistance in identifying specific taxa. We also thank assistants Alexandra Johnson and Janelle Goodine, and volunteers Connor Nelson, Olivia DeBourcier, Zachary Roote, Ferf Brownoff, Sara Peterson, Olivia Hrehoruk, Rykkar Jackson, and Victoria Dubord for their support on this project. We thank Edwin and Ruth Mattheis for donating the Mattheis Research Ranch to the University of Alberta, and Marcel Busz and Lisa Raatz for assistance at the ranch. We also thank the Eastern Irrigation District for allowing the use of their land. Honey bees were generously supplied by Pankratz Beekeeping.

## Author Contributions

**Conceptualization:** Sydney H. Worthy, John H. Acorn, Carol M. Frost.

**Data curation:** Sydney H. Worthy.

**Formal analysis:** Sydney H. Worthy.

**Funding acquisition:** Sydney H. Worthy, Carol M. Frost.

**Investigation:** Sydney H. Worthy, John H. Acorn.

**Methodology:** Sydney H. Worthy.

**Project administration:** Sydney H. Worthy.

**Resources:** Carol M. Frost.

**Supervision:** John H. Acorn, Carol M. Frost.

**Visualization:** Sydney H. Worthy.

**Writing – original draft:** Sydney H. Worthy.

**Writing – review & editing:** Sydney H. Worthy, John H. Acorn, Carol M. Frost.

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
