## [Decision Letter · Decision Letter 0]

6 Mar 2024

PONE-D-23-43148Biodiversity measures of a grassland plant-pollinator community are resilient to the introduction of honey bees (Apis mellifera)PLOS ONE

Dear Dr. Worthy,

Thank you for submitting your manuscript to PLOS ONE. After careful consideration, we feel that it has merit but does not fully meet PLOS ONE’s publication criteria as it currently stands. Therefore, we invite you to submit a revised version of the manuscript that addresses the points raised during the review process.

We look forward to receiving your revised manuscript.

Kind regards,

Javaid Iqbal, PhD

Academic Editor

PLOS ONE

Journal Requirements:

3. We note that Figures 1, S1, S5 and Tables S1, S2, S5, S6 in your submission contain copyrighted images. All PLOS content is published under the Creative Commons Attribution License (CC BY 4.0), which means that the manuscript, images, and Supporting Information files will be freely available online, and any third party is permitted to access, download, copy, distribute, and use these materials in any way, even commercially, with proper attribution. For more information, see our copyright guidelines: http://journals.plos.org/plosone/s/licenses-and-copyright.

We require you to either present written permission from the copyright holder to publish these figures specifically under the CC BY 4.0 license, or remove the figures from your submission:

a. You may seek permission from the original copyright holder of Figures 1, S1, S5 and Tables S1, S2, S5, S6 to publish the content specifically under the CC BY 4.0 license. 

Reviewers' comments:

Reviewer's Responses to Questions

**Comments to the Author**

1. Is the manuscript technically sound, and do the data support the conclusions?

Reviewer #1: Yes

Reviewer #2: No

Reviewer #3: Partly

2. Has the statistical analysis been performed appropriately and rigorously? 

Reviewer #1: Yes

Reviewer #2: No

Reviewer #3: No

3. Have the authors made all data underlying the findings in their manuscript fully available?

Reviewer #1: Yes

Reviewer #2: Yes

Reviewer #3: Yes

4. Is the manuscript presented in an intelligible fashion and written in standard English?

Reviewer #1: Yes

Reviewer #2: Yes

Reviewer #3: Yes

5. Review Comments to the Author

Reviewer #1: The article entitled "Biodiversity measures of a grassland plant-pollinator community are resilient to the introduction of honey bees (Apis mellifera)" evaluates the effect of honey bee (Apis mellifera) density on diversity, abundance, species composition of wild pollinators and on the diversity of plant-wild pollinator interactions in the Canadian prairies. Due to habitat loss, the negative effects of pesticides, the increase in bee diseases, etc., wild pollinators are at serious risk of declining populations and diversity, so studies to better understand the effects and risks to wild pollinators can contribute to their conservation at a global level.

These risks are also suffered by honey bees, which are managed by beekeepers who are forced to introduce them into natural areas in search of pollen resources for food and production. The increase of honey bee hives in natural areas and their effects on wild pollinators is a topic of great importance at present, about which many factors are still unknown, so studies such as the present work are necessary to be able to make management plans on the introduction of honey bee hives in natural areas.

The article is well written and organised. The introduction sets out the background to the work, the objectives in an appropriate manner. The methodology used is well explained and detailed.

The objectives of this study are to:

1. determine whether honey bee abundance impacts the species richness, diversity, abundance, or species composition of wild pollinators;

2. determine whether honey bee abundance impacts the richness and diversity of plant pollinator interactions;

3. determine whether honey bee abundance affects the richness, diversity or abundance of certain higher taxa (bees, beetles, flies, and butterflies) more than others;

4. determine whether effects of honey bees are greater when honey bee colony sizes are largest, and whether this differs for different higher taxa.

In general, the objectives have been completed correctly, with an adequate methodology and with a correct exposition of the results. However, the fourth objective (determine whether effects of honey bees are greater when honey bee colony sizes are larger, and whether this differs for different higher taxa), is the one that could be methodologically improved by including some more data. To know more exactly the density of honey bees in the territory, the census of hives in the vicinity should be known and taken into account. As well as the state of development of the honey bee hives introduced in the study, i.e., data on the increase of brood frames, percentage of workers per frame, weight of the hives, production at the end of the season, etc., should be included in order to have objective indexes of the growth of the hive and be sure that this leads to an increase in the density of honey bees throughout the season, since any sanitary, nutritional or management problem can produce a population decrease and the death of the hive. It is recommended that such data be included in the analysis.

Stan Chabert, Fabrice Requier, Joël Chadoeuf, Laurent Guilbaud, Nicolas Morison, Bernard E. Vaissière. 2021. Rapid measurement of the adult worker population size in honey bees. Ecological Indicators 122:107313, DOI: 10.1016/j.ecolind.2020.107313.

A question for the authors:

- What is the average size of an apiary in Canada, as professional apiaries between 16-48 hives seem to me to be of a small size, so the effects of large apiaries (larger than 150 hives) have not been studied in the present work and should be mentioned in the discussion and conclusion. The present work is valid only for small size apiaries.

Other considerations:

- In Figure 2, there is an error in the name: (8) Full-season Butterflies.

- In figure 3, indicate in the graph the regression line data, confidence intervals, etc...

But overall, it is a suitable article for publication. I therefore recommend "major revision".

Reviewer #2: In this manuscript Worthy et al. aim to investigate the effects of honey bees densities on a wild pollinator community of a Canadian grassland. Although the dataset is interesting, the paper does not properly justify why this new exploration of the data is needed considering the already published paper "Honey bees (Apis mellifera) modify plant-pollinator network structure, but do not alter wild species’ interactions" by the same authors. In lines 83-87 the authors write a few lines in this regard but these are not enough to justify a new publication. The methods have a high overlap with the previously cited publication, including several identical paragraphs. On top of this there are a number of concerns and unclarities in the methods and statistics. Because of the above I consider that in its current form the manuscript is unsuitable for publication.

Reviewer #3: The authors present a comprehensive study on the impact of introduced honeybees on the plant-pollinator community by analyzing several diversity metrics on different wild pollinators. The result obtained adds an interesting point of view and feeds the scientific debate on the effect of honeybees. I think it is important to publish this type of articles with negative or null results for scientific progress. However, in the present form, the manuscript has some shortcomings that prevent its publication.

The main objection is about the methodological approach used. Since the abundance and richness of floral resources is probably playing a relevant role as a predictor variable of pollinator diversity metrics, I do not know if the most appropriate methodological approach is to start analyzing the effect of honeybee abundance on SLRs and then explore with MR, the effect of other variables as predictor variable (with floral resources among them). Given honeybees' capacity to exploit high-abundance floral resources because of their ability to recruit nest-mates, both variables could be correlated. I believe that a more proper approach would be to first analyze the correlation between the different variables and then explore with MR different predictor variables avoiding the correlated ones (using the VIF).

I have listed some comments and suggestion below

Introduction

Line 46: “where they are not native”. Also, hb are widely used within its native range. This sentence may lead the reader to interpret impacts occur only in areas where it has been introduced and not in its native range.

Line 48-50: rephrase.

Material and Methods

The three clusters are 3 km apart, which is closer than the maximum distance for each cluster (5km). Then make it clear the distances among 5000m hives, and how you deal with this effect. For example, E5000 is closer to A500 or A100 than to E500 or E100

As stated in the discussion maybe is better to describe in this section that there were no known apiaries within 19 km of the sampling area, that could have an effect on the experimental design (mainly on 5000m sites).

Why haven’t you analyzed the full dataset using both hand-caught and pan-trapped pollinators? The same in pan-trapped dataset: why do not use a full-season and mid-season with all-taxa dataset?

Results

I believe that figure 2 can be eliminated as it is not providing relevant information and can be replaced by another figure that provides an understanding of the results.

I have missed in the section "Effect of hb on pollinator diversity metrics" a first descriptive part detailing the richness and abundance of the different groups in the 3 distance categories (100,500 and 5000). Perhaps to avoid a greater number of tables, it can be included in table S3 or S5,or S7 summarized by group/family, so that the reader can check the raw data.

In the fig S6 it seems that in the Mid-season dataset there seem to be an effect on pollinator richness hand-caught between the three distances. Have you explored this?

6. PLOS authors have the option to publish the peer review history of their article (what does this mean?). If published, this will include your full peer review and any attached files.

Reviewer #1: No

Reviewer #2: No

Reviewer #3: No

---

## [Author Response · Author response to Decision Letter 0]

3 May 2024

Dear Dr. Iqbal,

Thank you for allowing us the opportunity to re-submit our manuscript to PLOS ONE. We have revised our manuscript “Biodiversity measures of a grassland plant-pollinator community are resilient to the introduction of honey bees (Apis mellifera)” in light of the reviewers’ comments, and we believe that the manuscript is improved as a result. We are grateful for these comments and have responded specifically to each suggestion. We have marked our responses with ‘>’. Additionally, we have addressed the journal requirements listed in the decision letter and added a full ethics statement in the Methods section, as well as have addressed the copyright issues regarding the figures and tables.

We thank you for your time and consideration.

Yours sincerely,

Sydney Worthy

John Acorn

Carol Frost

Reviewer #1 (Comments to Author): 

The article entitled "Biodiversity measures of a grassland plant-pollinator community are resilient to the introduction of honey bees (Apis mellifera)" evaluates the effect of honey bee (Apis mellifera) density on diversity, abundance, species composition of wild pollinators and on the diversity of plant-wild pollinator interactions in the Canadian prairies. Due to habitat loss, the negative effects of pesticides, the increase in bee diseases, etc., wild pollinators are at serious risk of declining populations and diversity, so studies to better understand the effects and risks to wild pollinators can contribute to their conservation at a global level.

These risks are also suffered by honey bees, which are managed by beekeepers who are forced to introduce them into natural areas in search of pollen resources for food and production. The increase of honey bee hives in natural areas and their effects on wild pollinators is a topic of great importance at present, about which many factors are still unknown, so studies such as the present work are necessary to be able to make management plans on the introduction of honey bee hives in natural areas.

The article is well written and organised. The introduction sets out the background to the work, the objectives in an appropriate manner. The methodology used is well explained and detailed.

> We thank the reviewer for these positive comments.

In general, the objectives have been completed correctly, with an adequate methodology and with a correct exposition of the results. However, the fourth objective (determine whether effects of honey bees are greater when honey bee colony sizes are larger, and whether this differs for different higher taxa), is the one that could be methodologically improved by including some more data. To know more exactly the density of honey bees in the territory, the census of hives in the vicinity should be known and taken into account. 

>We thank the reviewer for this idea and agree it would be ideal to have greater certainty on the exact number of hives in the region of the Mattheis Research Ranch. However, we did obtain and present this information in as much detail as possible. We have now made this information more prominent, however, by moving it to the methods. We now report on L126-128, “the Eastern Irrigation District, owner of the land surrounding the Mattheis Research Ranch, reported no known apiaries within 19 km of the ranch’s boundary line within the study year or the previous year”. 

There was also a natural “barrier” of the Red Deer River, which borders the northern and eastern edges of the Ranch and has a tributary on the southern edge. The river is approximately 225 m wide on average and while we cannot be sure of the extent it deters insect travel, there is evidence that rivers discourage flying insects (Sciarretta & Trematerra 2006, Journal of Applied Entomology, Volume 130, Issue 2, Pages 73-83) even if not insurmountable (Zurbuchen et al., 2010, Journal of Animal Ecology, Volume 79, Issue 3, Pages 674-681). 

Additionally, because we drove to 20 km from the Mattheis Research Ranch in all driveable directions and did not observe any apiaries from the road, we feel that we confirmed adequately (and as well as it was possible to do) that large-scale commercialized honey bee operations were not close enough to the study site to invalidate our design. Feral hives, and illegal, unregistered hives, may have been possible, as we mention on L193. 

However, information on the exact density of honey bee colonies in the region would be more important to the validity of our study if we had used distances from hives as the predictor variable representing honey bee density. However, we did not do this, partly because we did not have perfect certainty as to whether feral or unregistered hives could have been present on the landscape. Rather, we used the number of hand-caught honey bees on our transects. Thus, we used an empirically-measured estimate of honey bee density at the transects where we were measuring wild pollinator abundance, diversity and interactions. This is because honey bee foragers share information about resource location and quality, such that we would not expect distance alone to explain honey bee activity around flowers, but also floral patch quality. As such, we believe that our choice of predictor variable negates this concern, especially as it allowed us to realize that one of our originally selected 5000-m sites was unexpectedly receiving many visits from honey bees, and to relocate it to another location, where there was negligible honey bee activity. If unknown, additional honey bee colonies had contributed interactions to our transects, we should have detected them during sampling as we did at the transect F5000.

As well as the state of development of the honey bee hives introduced in the study, i.e., data on the increase of brood frames, percentage of workers per frame, weight of the hives, production at the end of the season, etc., should be included in order to have objective indexes of the growth of the hive and be sure that this leads to an increase in the density of honey bees throughout the season, since any sanitary, nutritional or management problem can produce a population decrease and the death of the hive. It is recommended that such data be included in the analysis.

>We agree with the reviewer that this information would support the data; however, it must be considered that this information is now impossible to retrieve. During the study, we confirmed bi-weekly, with the apiarist, that the hives were healthy at the study sites. Additionally, we visited the hives weekly to ensure that no animals had disturbed them, that swarms were not starting, and that the hives appeared active. We have now added: “The hives were assessed bi-weekly by the apiarist for health and productivity throughout the season. By late summer, all hives grew significantly and had 4 to 6 supers added per hive.” (L120-122).

A question for the authors:

What is the average size of an apiary in Canada, as professional apiaries between 16-48 hives seem to me to be of a small size, so the effects of large apiaries (larger than 150 hives) have not been studied in the present work and should be mentioned in the discussion and conclusion. The present work is valid only for small size apiaries.

>We agree with the reviewer that this would be a small-scale operation and thus have adjusted the language to suit this change (L120).

Other considerations:

In Figure 2, there is an error in the name: (8) Full-season Butterflies.

>We believe this mark may have appeared while converting the image file in the PLOS ONE PACE image conversion services. We will ensure it is not present upon resubmission. 

- In figure 3, indicate in the graph the regression line data, confidence intervals, etc...

>We have adjusted Figure 3 to include regression line data and confidence intervals, as suggested.

Reviewer #2 (Comments to Author): 

In this manuscript Worthy et al. aim to investigate the effects of honey bees densities on a wild pollinator community of a Canadian grassland. Although the dataset is interesting, the paper does not properly justify why this new exploration of the data is needed considering the already published paper "Honey bees (Apis mellifera) modify plant-pollinator network structure, but do not alter wild species’ interactions" by the same authors. In lines 83-87 the authors write a few lines in this regard but these are not enough to justify a new publication.

> We appreciate this concern, but the current manuscript reports a completely different analysis of the same data as Worthy et al. 2023, and also adds an entirely new dataset: the pan-trapped dataset. Worthy et al. 2023 modelled the plant-pollinator interactions measured as described here as a network (which we do not do here), and then tested whether the structural metrics that described this network changed as honey bee abundance increased. However, reviewers of that manuscript commented that it would be possible for pollinators to disappear or change in relative abundance (i.e., pollinator richness or diversity could decrease), or for certain interactions to disappear (i.e. interaction richness and diversity could decrease) even without significant changes to the network metrics reported. This is true, as network metrics are not fully dependent on biodiversity metrics. (For a striking example of this, see Tylianakis et al. 2007. Nature 445: 202-205.) As such, the biodiversity results reported here are a separate and equally important part of the story that was not reported in Worthy et al., 2023, simply because that paper already included a huge number of models, and it was not possible to include everything in one reasonably-lengthed paper.

To explain it another way, Worthy et al., 2023 reports effects of honey bee abundance on functional characteristics of the entire pollinator community, whereas the current manuscript reports on effects of honey bee abundance on biodiversity of the wild pollinator community. 

We agree strongly with Reviewer 3 that publication of these “negative or null” results on this topic are very important, because there are so many studies reporting negative effects honey bees on exactly the metrics reported in the current manuscript. It is not possible to tell from reading Worthy et al., 2023 that the wild pollinator diversity and interaction diversity (altogether or for higher taxa separately) did not change as honey bee abundance increased, which is what we report here.

This said, the reviewer’s comment is useful, because it is important that the distinction between the two studies is clear, and we have improved our justification of the need for this further analysis of some of the data used in Worthy et al., 2023 (L112-116). 

The methods have a high overlap with the previously cited publication, including several identical paragraphs. On top of this there are a number of concerns and unclarities in the methods and statistics. Because of the above I consider that in its current form the manuscript is unsuitable for publication.

>We have now modified the methods section of our paper to address the concern of high overlap. The reviewer did not explain what they found unclear, but we have edited the entire section for clarity.

Reviewer #3:

The authors present a comprehensive study on the impact of introduced honeybees on the plant-pollinator community by analyzing several diversity metrics on different wild pollinators. The result obtained adds an interesting point of view and feeds the scientific debate on the effect of honeybees. I think it is important to publish this type of articles with negative or null results for scientific progress. However, in the present form, the manuscript has some shortcomings that prevent its publication.

> We thank the reviewer for these positive comments and agree that negative/null results are important.

The main objection is about the methodological approach used. Since the abundance and richness of floral resources is probably playing a relevant role as a predictor variable of pollinator diversity metrics, I do not know if the most appropriate methodological approach is to start analyzing the effect of honeybee abundance on SLRs and then explore with MR, the effect of other variables as predictor variable (with floral resources among them). Given honeybees' capacity to exploit high-abundance floral resources because of their ability to recruit nest-mates, both variables could be correlated. I believe that a more proper approach would be to first analyze the correlation between the different variables and then explore with MR different predictor variables avoiding the correlated ones (using the VIF).

> We respectfully disagree, which we explain below, but we have still calculated VIF for interest (L281-285), and have added them to the supplement to show that they are not high enough to affect our results or interpretations.

We agree with the reviewer that in some applications of multiple regression, VIF are useful, because they show the degree to which each predictor variable is linearly correlated with the full set of other predictor variables in the model. If the VIF for a given predictor variable is high, the variance associated with that predictor variable in the model analysis is high, and therefore the P-value has substantially less likelihood of being significant in the multiple regression (MR) than if that predictor variable had been the only predictor variable in a simple linear regression (SLR) with the same response variable. 

VIF are useful for knowing if the other covariates in the model are masking the effect of a focal predictor variable on the response variable. We have fully addressed this issue using another approach, which we believe is superior, because it doesn’t involve leaving out any predictor variables that we want to control for, which is what the reviewer is suggesting, should one or more of these variables create VIF that are “too high”. While we acknowledge that some statistics textbooks advocate the blanket application of VIF to multiple regression analysis, the VIF literature itself advocates more thoughtful use of VIF, including the statement they are not always useful, depending on the purpose of the analysis (e.g., O’Brien, R.M. 2007. A caution regarding rules of thumb for variance inflation factors. Quality & Quantity 41: 673-690), which some statistics textbooks also state (e.g., Jones, Hardin, & Crawley. 2023. “The R Book”. Third Edition. John Wiley & Sons Ltd. NJ, USA).

The problem that the reviewer mentions of potential multicollinearity between our predictor variable of interest (honey bee abundance) and the predictor variables that we included in the MR models as controls (flower abundance, flower species richness, and collection effort), is exactly the reason why we used the two-step approach of first running an SLR testing the effect of honey bee abundance alone on each response variable (not controlling for other potentially correlated variables that might also affect the response variable), and then running an MR model that includes all these other variables. The SLR results show the maximum possible effect size of honey bee abundance on each response variable (Morrissey & Ruxton, 2018). These SLR models are therefore effectively the least conservative assessment of the effect of honey bee abundance on each response variable. They are less conservative than the reviewer’s suggestion of using VIF to select some additional variables to put in the model. We wish to include this least conservative view of the potential effect of honey bee abundance on each response variable, because we believe it is important. This is because, in MR models, it is impossible to distinguish where the true causation lies when multiple predictor variables are correlated but significantly related to the response variable (Morrissey & Ruxton, 2018). As such, if we used VIF to exclude variables that had high collinearity with one or more other variables, we could be excluding the causative variable and keeping the non-causative but correlated variable.

Because of the inability of multiple regression to distinguish correlation and causation, we believe that the best we can do at assessing effects of honey b

---

## [Decision Letter · Decision Letter 1]

31 Jul 2024

PONE-D-23-43148R1Biodiversity measures of a grassland plant-pollinator community are resilient to the introduction of honey bees (Apis mellifera)PLOS ONE

Dear Dr. Worthy,

Thank you for submitting your manuscript to PLOS ONE. After careful consideration, we feel that it has merit but does not fully meet PLOS ONE’s publication criteria as it currently stands. Therefore, we invite you to submit a revised version of the manuscript that addresses the points raised during the review process.

We look forward to receiving your revised manuscript.

Kind regards,

Javaid Iqbal, PhD

Academic Editor

PLOS ONE

Journal Requirements:

Reviewers' comments:

Reviewer's Responses to Questions

**Comments to the Author**

1. If the authors have adequately addressed your comments raised in a previous round of review and you feel that this manuscript is now acceptable for publication, you may indicate that here to bypass the “Comments to the Author” section, enter your conflict of interest statement in the “Confidential to Editor” section, and submit your "Accept" recommendation.

Reviewer #1: All comments have been addressed

Reviewer #4: All comments have been addressed

Reviewer #5: (No Response)

2. Is the manuscript technically sound, and do the data support the conclusions?

Reviewer #1: Yes

Reviewer #4: Yes

Reviewer #5: Yes

3. Has the statistical analysis been performed appropriately and rigorously? 

Reviewer #1: Yes

Reviewer #4: Yes

Reviewer #5: Yes

4. Have the authors made all data underlying the findings in their manuscript fully available?

Reviewer #1: Yes

Reviewer #4: Yes

Reviewer #5: Yes

5. Is the manuscript presented in an intelligible fashion and written in standard English?

Reviewer #1: Yes

Reviewer #4: Yes

Reviewer #5: Yes

6. Review Comments to the Author

**Reviewer #1:** The authors have improved the manuscript and included the reviewers' corrections, so that it is now suitable for publication.

**Reviewer #4: **Minor linguistic revision.

**Reviewer #5:** Two minor corrections:

1- Adding keywords to the MS

2- Table 2 caption the word contained is repeated twice

7. PLOS authors have the option to publish the peer review history of their article (what does this mean?). If published, this will include your full peer review and any attached files.

Reviewer #1: No

Reviewer #4: **Yes: **Ayman A. Owayss

Reviewer #5: No

---

## [Author Response · Author response to Decision Letter 1]

6 Aug 2024

Dear Dr. Iqbal,

Thank you for allowing us the opportunity to re-submit our manuscript to PLOS ONE. We have revised our manuscript “Biodiversity measures of a grassland plant-pollinator community are resilient to the introduction of honey bees (Apis mellifera)” in light of the reviewers’ comments. We are grateful for these comments and have responded specifically to each suggestion. We have marked our responses with ‘>’. 

We thank you for your time and consideration.

Yours sincerely,

Sydney Worthy

John Acorn

Carol Frost

Reviewer #1: The authors have improved the manuscript and included the reviewers' corrections, so that it is now suitable for publication.

> We thank the reviewer for this positive review.

Reviewer #4: Minor linguistic revision.

> Since no specific section needing revision was mentioned, we reread the whole manuscript focusing on linguistic clarity. In the end, we revised only the point mentioned below.

Reviewer #5: Two minor corrections:

1- Adding keywords to the MS

2- Table 2 caption the word contained is repeated twice.

>We have modified the keywords in editorial manager to better reflect the language used in the manuscript by changing the keyword “mixedgrass prairie” to “prairie”.

>We have removed the repeated word “contained” from the caption of Table 2 (L377).

---

## [Decision Letter · Decision Letter 2]

22 Aug 2024

Biodiversity measures of a grassland plant-pollinator community are resilient to the introduction of honey bees (Apis mellifera)

PONE-D-23-43148R2

Dear Dr. Worthy,

We’re pleased to inform you that your manuscript has been judged scientifically suitable for publication and will be formally accepted for publication once it meets all outstanding technical requirements.

Kind regards,

Vicente Martínez López

Academic Editor

PLOS ONE

Additional Editor Comments (optional):

Reviewers' comments:

Reviewer's Responses to Questions

**Comments to the Author**

1. If the authors have adequately addressed your comments raised in a previous round of review and you feel that this manuscript is now acceptable for publication, you may indicate that here to bypass the “Comments to the Author” section, enter your conflict of interest statement in the “Confidential to Editor” section, and submit your "Accept" recommendation.

Reviewer #4: All comments have been addressed

Reviewer #5: All comments have been addressed

2. Is the manuscript technically sound, and do the data support the conclusions?

Reviewer #4: Yes

Reviewer #5: Yes

3. Has the statistical analysis been performed appropriately and rigorously? 

Reviewer #4: Yes

Reviewer #5: Yes

4. Have the authors made all data underlying the findings in their manuscript fully available?

Reviewer #4: Yes

Reviewer #5: Yes

5. Is the manuscript presented in an intelligible fashion and written in standard English?

Reviewer #4: Yes

Reviewer #5: Yes

6. Review Comments to the Author

Reviewer #4: None…

................…......

.........................

.........................

.........................

.........................

Reviewer #5: Thanks for addressing all reviewers comments and suggestions and I don't have any further comments for authors.

7. PLOS authors have the option to publish the peer review history of their article (what does this mean?). If published, this will include your full peer review and any attached files.

Reviewer #4: **Yes: **Ayman A

Owayss

Reviewer #5: **Yes: **Mohamed Shebl Abd Elfattah

---

## [Editor Report · Acceptance letter]

29 Aug 2024

PONE-D-23-43148R2 

PLOS ONE

Dear Dr. Worthy, 

I'm pleased to inform you that your manuscript has been deemed suitable for publication in PLOS ONE. Congratulations! Your manuscript is now being handed over to our production team.

Kind regards, 

on behalf of

Dr. Vicente Martínez López 

Academic Editor

PLOS ONE